# Short Circuiting Transfer, Formation, and Microstructure of Ti-6Al-4V Alloy by External Longitudinal Magnetic Field Hybrid Metal Inert Gas Welding Additive Manufacturing

**DOI:** 10.3390/ma15217500

**Published:** 2022-10-26

**Authors:** Chao Shi, Hongwei Sun, Jiping Lu

**Affiliations:** 1School of Mechanical Engineering, Beijing Institute of Technology, Beijing 100081, China; 2Jiangsu Automation Research Institute, Lianyungang 222006, China

**Keywords:** Ti-6Al-4V alloy, additive manufacturing, external longitudinal magnetic field, microstructure

## Abstract

In this work, the external longitudinal magnetic field hybrid metal inert gas welding (M-MIG) additive manufacturing method is employed to produce the Ti-6Al-4V alloy part. The effect of process parameters on the droplet transfer formation and microstructure of the part was studied by a high-speed camera, optical microscope, and electron backscattered diffraction. The results showed that a typical short-circuiting transfer was obtained with the wire feeding speed of 2 m/min–4 m/min. An external longitudinal magnetic field had an obvious effect on the arc shape. The uniform formation of the deposition layer was obtained with the wire feeding speed of 4 m/min. The width of M-MIG deposition layer was greater than that of the MIG, and the width of M-MIG deposition layer was increased with the increase of the magnetic excitation current. The microstructure of the deposition layer was mainly comprised of acicular martensite α’ and massive martensite α_m_. In addition, the β grain size in the M-MIG was less than that of the MIG. The average microhardness of the MIG deposition layer was 281.6 HV, which was less than that of M-MIG.

## 1. Introduction

Titanium and titanium alloys are widely used in aerospace, chemical, and biomedical fields because of their high specific strength, good corrosion resistance, and good biocompatibility [1,2]. However, the deep processing of titanium is limited due to its easy oxidation, high melting point, and high price. However, additive manufacturing (AM), i.e., 3D printing technology, is a good choice, which can not only greatly improve the utilization rate of titanium alloy materials, but also realize the manufacturing of complex structural parts [3].

At present, titanium alloy additive manufacturing focuses on the selection of raw materials and heat sources [4,5]. The raw materials include spherical powders and wire materials. It is mature to use spherical powder for titanium alloy additive manufacturing [6]. However, the spherical powder of titanium alloy used in additive manufacturing generally requires high purity, good sphericity, and narrow particle size distribution. In addition, the powder is easily oxidized, which leads to the poor plasticity and toughness of parts. In order to develop the other additive manufacturing method, researchers try to use wire as a raw material for additive manufacturing. The heat sources, laser, electron beam, and arc plasma are usually used to melt wire for additive manufacturing [3]. Arc plasma-based additive manufacturing had been widely concerned for titanium alloy additive manufacturing owing to the advantages of high energy, fast scanning speed, and large melting pool volume [4].

Henckell et al. [7] indicated that the production of titanium aluminides’ additive manufacturing part was used by a hot-wire assisted GMAW process. Chen et al. [2] studied the geometric parameters and microstructure of Ti-6Al-4V parts during the CMT additive manufacturing. Murgau et al. reported the temperature and microstructure evolution in gas tungsten arc welding wire feed additive manufacturing of Ti-6Al-4V [8]. However, the coarse microstructure and significant anisotropy was easily obtained in the titanium alloy arc plasma-based additive manufacturing part. Some researchers proposed additional nucleation particles to improve the titanium alloy part [9,10,11]. Zhuo et al. [11] researched the influence of trace boron addition on microstructure and mechanical properties of TC11/TC17 in wire arc additive manufacturing. The results showed that the coarser α phases surrounding thin α phases were formed, and β grain size was refined obviously. External longitudinal magnetic field hybrid arc welding is also used to improve the microstructure and mechanical properties of welded joints under the action of electromagnetic oscillation [12]. Wang et al. [13] indicated that, by adding the magnetic field in the wire arc additive manufacturing, the microstructure and mechanical properties of Inconel 625 superalloy were improved.

It is clear that, in the previous studies, adding a magnetic field in the wire arc additive manufacturing has great potential to the improve titanium alloy arc plasma-based additive manufacturing part. However, there is little previous research about the effect of the magnetic field on the microstructure and mechanical properties of Ti alloy arc plasma-based additive manufacturing. In this work, the external longitudinal magnetic field hybrid gas metal arc welding additive manufacturing is used to produce the Ti-6Al-4V part. The effect of process parameters on the short-circuiting transfer, formation, and microstructure of Ti-6Al-4V alloy parts are studied in detail.

## 2. Experimental Material and Methods

In this work, welding wire of Ti-6Al-4V alloy (diameter of 1.2 mm) was used to produce the single deposition layer. The Ti-6Al-4V alloy plate with the dimension of 100 mm × 12 mm × 12 mm was selected as the base metal. The chemical compositions of Ti-6Al-4V alloy are contained Al element (5.5~6.8%), V element (3.5~4.5%), Fe element (≤0.30%), O element (≤0.20%), C element (≤0.10%), N element (≤0.05%), and Ti (balance).

The system of external longitudinal magnetic field hybrid gas metal arc welding additive manufacturing (M-GMAW-AM) is shown in Figure 1. The system mainly included a GMAW power source produced by Kemppi in Lahti, Finland, an external longitudinal magnetic field hybrid GMAW torch produced by Kemppi in Finland, and a high speed camera produced Optronis in Kehl, German. The coil turns of the longitudinal magnetic field are 140, and the magnetic excitation current is arranged from 1 A to 3 A. Under the action of the external longitudinal magnetic field, the metal transfer will be affected by the external electromagnetic force. The high speed camera is carried out to observe the process of short-circuiting transfer. The frame rate of 1000 fps and exposure time of 10 μs are employed at the capture parameters of the high-speed camera.

The fixed parameters are the voltage (19 V). The parameter variable contains wire feeding speed (2–4 m/min) and magnetic excitation current (1–3 A). The wire feeding speeds of 2 m/min, 3 m/min, and 4 m/min respectively correspond to the welding current of 150A, 165A, and 180A. The detail parameters of Ti-6Al-4V alloy in M-GMAW-AW are given in Table 1.

The single-layer deposition layer is used to study the relationship between the GMAW-AM parameter and the formation of the deposition layer. The surface morphology of the deposition layer is observed by a camera. The cross-section of the deposition layer is produced by using EDM wire cutting technology. The cross-section of the deposition layer will go through the ground and be polished. The cross-section is etched by using an alcohol solution containing 4% of nitric acid. An optical microscope (OM) produced by Shanghai Aolong Xingdi Testing Equipment Co., Ltd. in Shanghai, China is used to capture the images of a cross-section of the deposition layer. An HVS-1000 hardness tester produced by Shanghai Aolong Xingdi Testing Equipment Co., Ltd. in China is employed to measure the microhardness of deposition layer with the loading force of 200 g.

## 3. Results and Discussion

### 3.1. Short Circuiting Transfer

Figure 2 shows the images of short-circuiting transfer, when the wire feeding speed of 2 m/min is used. The typical features of short-circuiting transfer could be obtained in Figure 2. The short-circuiting transfer of GMAW is shown in Figure 2a. The phenomenon of short circuiting is found at 1462 ms and 1663 ms. The size of the droplet (1462 ms) was obviously larger than that of the droplet (1663 ms), which expressed that the stability of the deposition process was poor. At the stable arcing stage (1463 ms~1490 ms), the arc shape was biased to the left. However, when the external longitudinal magnetic field is added, the arc shape obviously rotates as shown in Figure 2b–d. The similar results had been reported that, under the action of axial electromagnetic force, the arc shape expressed periodic rotation [12]. Figure 2b shows the images of short-circuiting transfer in the M-GMAW with a magnetic excitation current of 1 A. The phenomenon of short circuiting is found at 17,850 ms and 18,103 ms, and the droplet size is less than that of Figure 2a. In addition, the droplet sizes at the 17,850 ms and 18,103 ms have a small difference. When the magnetic excitation current is increased to 2A, the droplet sizes also have a small difference. However, with the droplet sizes, an obvious difference is found when the magnetic excitation current of 3 A is used (Figure 2d).

Figure 3 shows the images of short-circuiting transfer, when the wire feeding speed of 3 m/min is used. Compared with Figure 2, the arc shape is expanded. This is because the current was increased with the increase of the wire feeding speed. In this work, the exposure time of high-speed camera was set as 10 μs. When the larger current was used, the larger arc shape would be observed. The rotation of the arc shape is also obtained by adding an external longitudinal magnetic field, as shown in Figure 3b–d. Although the size difference between the different droplet was found, the difference had obviously been reduced compared with that of Figure 2. The size of a single droplet (Figure 3) was also reduced compared with that of Figure 2.

Figure 4 shows the images of short-circuiting transfer, when the wire feeding speed of 4 m/min is used. A stable short-circuiting transfer process is obtained because the size difference of different droplets basically disappeared. Ref. [14] indicated that the stability of short-circuiting transfer process was mainly affected by the current and voltage. In this work, the voltage remained constant, and the current was increased with the increase of wire feeding speed. Normally, the higher or lesser current could not obtain a stable short-circuiting transfer process due to the complex force conditions of droplet transfer. The arc length was reduced compared with Figure 2 and Figure 3. In addition, the rotation of arc shape was also obtained by adding an external longitudinal magnetic field.

From Figure 2, Figure 3 and Figure 4, it can be concluded that, with the different wire feeding speeds, the change tendency of short-circuiting transfer was the same. However, the arc shape and droplet size had an obvious difference. For the arc shape, the larger area of arc shape obtained by the wire feeding speed of 2 m/min is due to the instability of the welding process. When the wire feeding speed increased, the stability of the welding process was improved, which results in the arc shape having no obvious change. In addition, the droplet size was reduced by the increase of wire feeding speed.

Figure 5 shows the cycles of short-circuiting transfer with the different wire feeding speed. The cycles of short-circuiting transfer obtained by calculation the average of three cycles. It can be seen from Figure 5 that the cycles of short-circuiting transfer are reduced with the increase of wire feeding speed. The cycles of short-circuiting transfer range from 200 ms to 300 ms when the wire feeding speed of 2 m/min is used. The cycles of short-circuiting transfer with the wire feeding speed of 3 m/min range from 120 ms to 220 ms. When the wire feeding speed is increased to 4 m/min, the cycles of short-circuiting transfer are about 80 ms. It can also be seen that the cycles of short-circuiting transfer of MIG were less than that of M-MIG when the wire feeding speed of 2 m/min and 3 m/min was used, and the different processes have a larger difference in the cycles of the same wire feeding speed. When the wire feeding speed of 4 m/min was used, the cycles of different processes had little difference.

Normally, surface tension can hinder the droplet transfer and keep droplet morphology [14]. During the short-circuiting transfer, when the combined force of gravity and other forces exceed the surface tension, the droplet can transfer to a welding pool. The surface tension can be expressed as *F_γ_* = 2πRγ, the R is the radius of welding wire, and the γ is the surface tension coefficient. In this work, the radius of welding wire and surface tension coefficient is the same owing to the same welding wire during the GMAW and M-MIG. Therefore, the hindering force of droplet transfer is the same in the GMAW and M-MIG. The different gravity and other forces are the main causes of the change of droplet transfer, as shown in Figure 5. It can be found that the short-circuiting transfer cycles in the M-MIG were larger than that of MIG when the same wire feeding speed was employed. Ref. [12] indicated that the charged particles rotated with a higher speed in the M-MIG compared with that of MIG. The motion path of charged particles in the M-MIG was increased compared with that of MIG. The arc had an increased loss of energy of heating and motion. This is meant that, when the welding voltage is constant, the resistance of the arc increases, and the welding current and arc temperature decrease. Therefore, short-circuiting transfer cycles in the M-MIG were less than that of the MIG.

### 3.2. Formation

Figure 6 shows the images of surface morphology of the deposition layer with the different short-circuiting transfers. The *I_M_* expresses the magnetic excitation current. Figure 6a,b shows the surface morphology of the deposition layer with the wire feeding speed of 2 m/min. The deposition layers with different processes can not be forming, which consists of the discontinuous deposition layer. It can be concluded that the non-forming of deposition layer may be affected by the instability of droplet transfer (Figure 2). The discontinuous flow of molten metal was the direct reason for the discontinuous deposition layer. When the wire feeding speed is increased to 3 m/min, the forming of the deposition layer obviously changed as shown in Figure 6c,d; however, there are some spatters on the surface of deposition layer due to the change of droplet transfer path caused by a magnetic field. In addition, when the magnetic excitation current is increased to above 2A, the forming of the deposition layer is poor. This is because a larger electromagnetic force obtained by an electromagnet could adversely affect the droplet transfer, such as a great change of arc shape and spatter. When the wire feeding speed of 4 m/min is used, the uniform forming of the deposition layer is obtained as shown in Figure 6e,f. From Figure 4, it also can be found that the stable droplet transfer was obtained with the wire feeding speed of 4 m/min. Therefore, the stable droplet transfer was the key to getting the uniform forming of the deposition layer. 

Figure 7 shows the images of cross-sectional morphology of the deposition layer with the wire feeding speed of 4 m/min. Figure 7a shows the cross-sectional morphology of the MIG deposition layer. Figure 7b–d shows the cross-sectional morphology of the M-MIG deposition layer with the different magnetic excitation current. It can be found that the grain size in the M-MIG is refined obviously compared with that of the MIG. During the M-MIG solidification process, the molten pool was further stirred by the external longitudinal magnetic field, which led to dendrite fragmentation [15]. Figure 7e shows the geometric parameter of the cross-sectional morphology of the deposition layer. The width and height of MIG deposition layer are respectively 5.6 mm and 5.9 mm. When the external longitudinal magnetic field is added, the width of the deposition layer is increased to above 10 mm. In addition, the width was increased with the increase of the magnetic excitation current. However, the height of the M-MIG deposition layer is less than that of MIG. Under the action of the external longitudinal magnetic field, the rotation of the arc shape was obtained to enhance the spreading of the molten pool [16], and the width of the deposition layer obtained in the M-MIG. 

### 3.3. Microstructure and Microhardness

Figure 8 shows the images of phase distribution of the cross-section of the deposition layer obtained by using wire feeding speed of 4 m/min. α phase is shown in red and β phase is shown in green. Figure 8a shows the phase distribution of the cross-section of the deposition layer in the MIG. The content of the α phase is 0.992, and the content of β phase is 0.008. Figure 8b shows the phase distribution of the cross-section of the deposition layer in the M-MIG with the magnetic excitation current of 1A. The content of α phase is 0.997, and the content of β phase is 0.003. When the magnetic excitation current is increased to 2A, the content of the α phase is 0.995, and the content of β phase is 0.005, as shown in Figure 8c. When the magnetic excitation current of 3A is used, the content of α phase is 0.995, and the content of β phase is 0.005, as shown in Figure 8d. Therefore, it could be concluded that the influence of the magnetic field on phase content was not obvious and had no regularity.

Figure 9 shows the images of grain morphology and misorientation of the cross-section of the deposition layer obtained by using wire feeding speed of 4 m/min. Figure 9a1 shows the grain morphology of the cross-section of the deposition layer. The microstructures are mainly comprised of acicular martensite α’ and massive martensite α_m_. Figure 8 also confirmed this phenomenon. The martensite morphology was mainly related to the cooling rate during the phase transition [17]. When the cooling rate was above 410 °C/s, the non-diffusible phase transition occurred, and acicular martensite α’ formed. When the cooling rate was less than 20 °C/s, the diffusible phase transition occurred, and Widmannstätten structure formed. When the cooling rate ranged from 20 °C/s to 410 °C/s, massive martensite formed. Moreover, increasing the cooling rate could refine the phase size. A comparison of the M-MIG with the MIG showed that the MIG grain size was greater than that of M-MIG. In addition, due to the rapid heat dissipation of the substrate, a high temperature gradient was easily obtained. Therefore, the coarse β grain of MIG grew along the <100> and obviously had a preferred orientation, as shown in Figure 9a1. However, when the M-MIG is used, the coarse β grain is obviously reduced and transformed into a fine equiaxed grain, as shown in Figure 9b1–d1.

Under the action of the external longitudinal magnetic field, the rotation of arc shape was obtained, which would fluctuate on the melt pool. At the same time, the magnetic field also affected the inside of the melt pool [15]. The liquid metal inside the molten pool had a complex flow under the action of arc pressure, electromagnetic force, and other forces. Under the action of strong stirring, a forced convection process would be formed in the weld pool. The high temperature liquid melting pool metal has changing flowing direction under the action of the varying Lorentz force, which created a uniform distribution of temperature field and solute field obtained in the molten pool. Therefore, the preferential growth of grain was limited, and the flushing action of the fluid would also promote the formation of new crystal nuclei, which could increase the number of crystal nuclei in the unit area of the liquid molten pool.

Figure 9a2 shows the misorientation of the cross-section of the deposition layer. The red region expresses the larger misorientation. The green region expresses the smaller misorientation. Figure 9a2 shows the misorientation of MIG. In this figure, a bigger red region is observed. β grain boundary is meant for larger misorientation. The larger misorientation in the M-MIG is less than that of MIG.

Figure 10 shows the average microhardness of the deposition layer obtained by using wire feeding speed of 4 m/min with different methods. The average microhardness of the deposition layer obtained by using MIG is 281.6HV. When the M-MIG with the magnetic excitation current of 1A is used, the average microhardness of the deposition layer is about 296 HV. When the M-MIG with the magnetic excitation current of 2 A and 3 A are used, the average microhardness of deposition layer is about 298.3 HV and 311.7 HV. It could be found that the average microhardness of M-MIG deposition layer was greater than that of the MIG, and the average microhardness of M-MIG deposition layer was increased with the increase of the magnetic excitation current. The causes for the change in microhardness could be summed up in two ways: first, the β grain was refined by using the magnetic field. According to the Hall–Petch formula, the microhardness was increased with the reduction of grain size [18]. In this work, the grain size in the M-MIG was obviously less than that of the MIG. Refinement grain was good for improving the microhardness. In addition, the martensite α could also determine the change tendency of microhardness. The microhardness was increased with the amount of acicular martensite α’ [10].

## 4. Conclusions

When the wire feeding speed of 2 m/min~4 m/min was used, the model of Ti alloy droplet transfer was typical of the short-circuiting transfer. Under the action of the external longitudinal magnetic field, the rotation of arc shape was obtained, and short-circuiting transfer cycles changed.The uniform formation of the deposition layer was obtained with the wire feeding speed of 4 m/min. The stable droplet transfer was the key to obtaining the uniform forming of the deposition layer. The external longitudinal magnetic field could enhance the spreading of the molten pool.The β grain size in the M-MIG was less than that of the MIG, which was caused by the electromagnetic stirring of the molten pool. The average microhardness of the MIG deposition layer was 281.6 HV, which was less than that of M-MIG.

## Figures and Tables

**Figure 1 materials-15-07500-f001:**
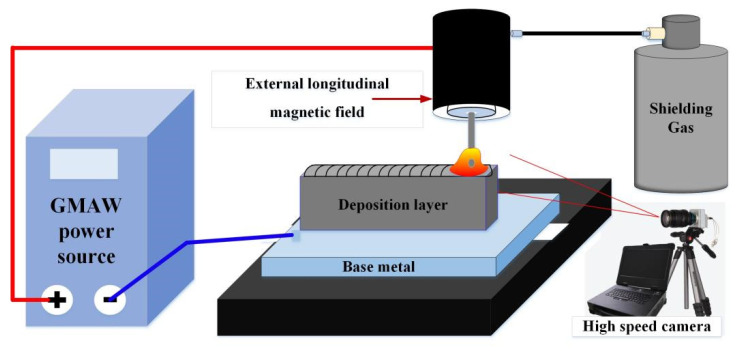
Schematic diagram of the M-GMAW-AM system.

**Figure 2 materials-15-07500-f002:**
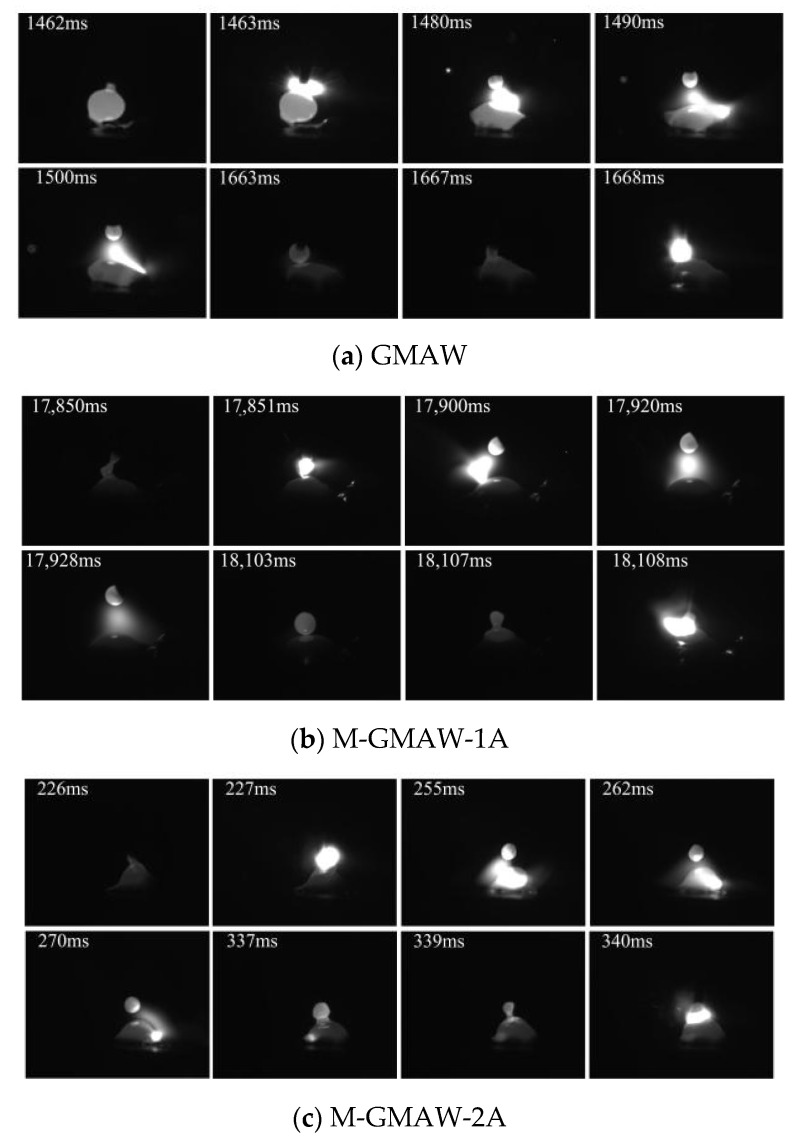
Wire feeding speed of 2 m/min.

**Figure 3 materials-15-07500-f003:**
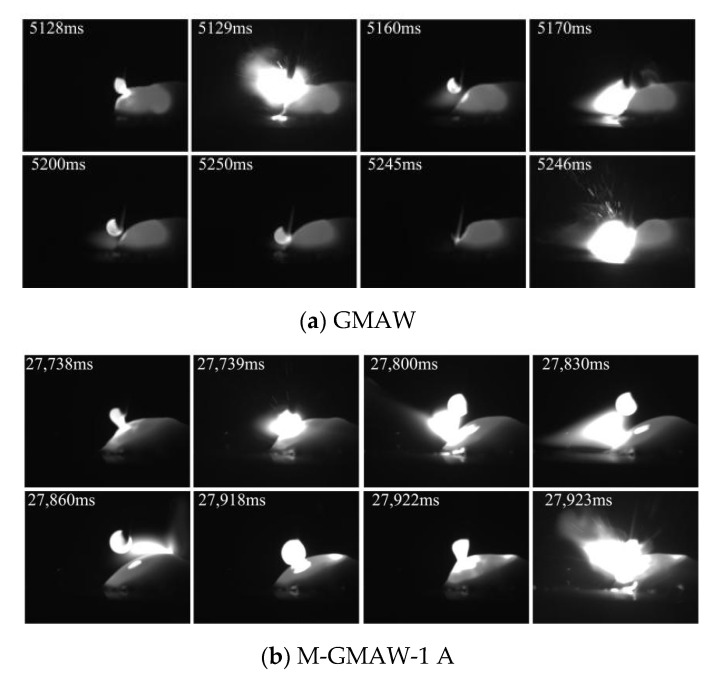
Wire feeding speed of 3 m/min.

**Figure 4 materials-15-07500-f004:**
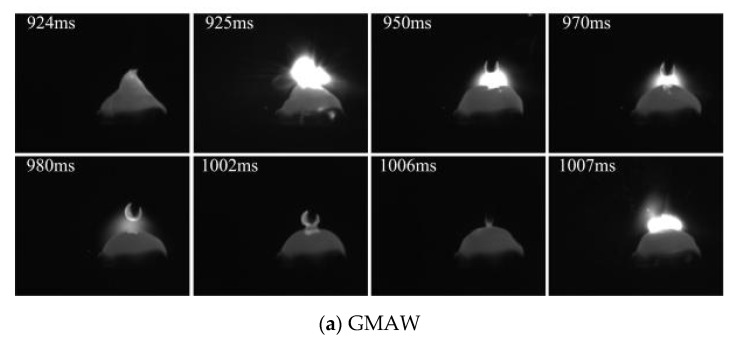
Wire feeding speed of 4 m/min.

**Figure 5 materials-15-07500-f005:**
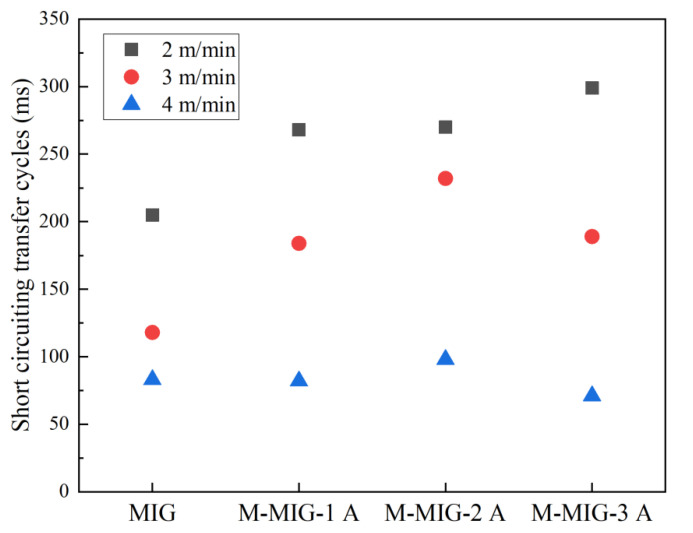
Short-circuiting transfer cycles with different wire feeding speed.

**Figure 6 materials-15-07500-f006:**
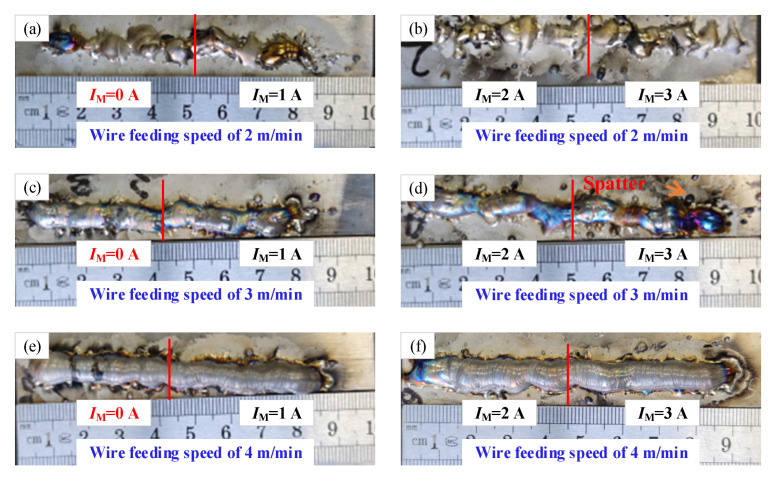
Surface morphology of deposition layer with the short-circuiting transfer.

**Figure 7 materials-15-07500-f007:**
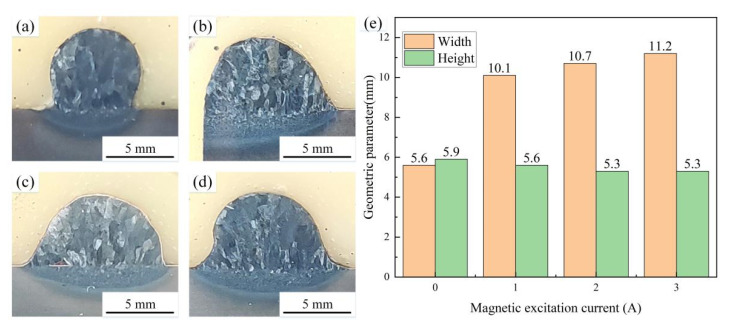
Cross-sectional morphology of the deposition layer with short-circuiting transfer. (Note: (**a**) MIG, (**b**) M-MIG with magnetic excitation current of 1A, (**c**) M-MIG with magnetic excitation current of 2A, (**d**) M-MIG with magnetic excitation current of 3A, (**e**) The geometric parameter of the cross-sectional morphology).

**Figure 8 materials-15-07500-f008:**
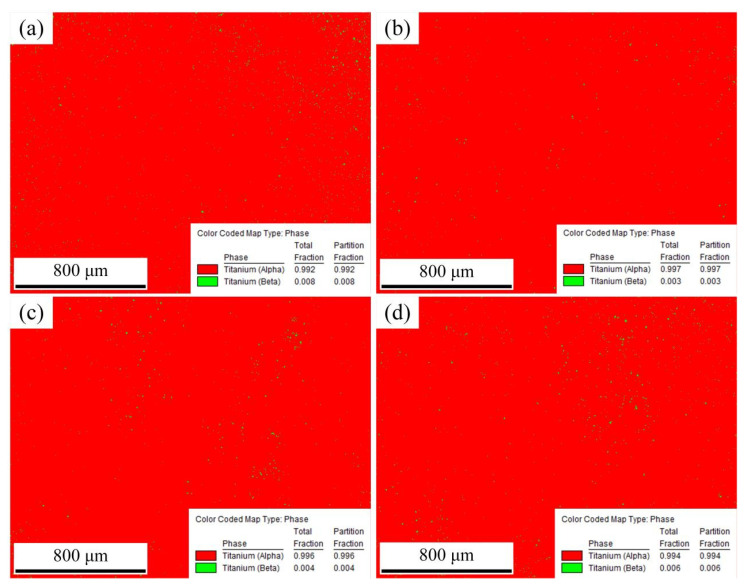
Phase distribution of the cross-section of the deposition layer obtained by using wire feeding speed of 4 m/min. (Note: (**a**) MIG, (**b**) M-MIG with magnetic excitation current of 1A, (**c**) M-MIG with magnetic excitation current of 2A, (**d**) M-MIG with magnetic excitation current of 3A).

**Figure 9 materials-15-07500-f009:**
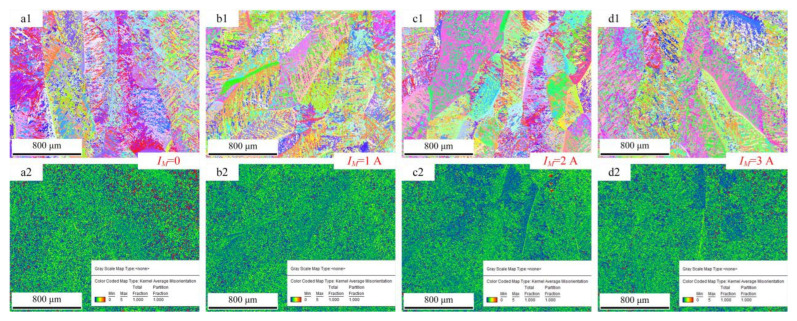
Grain morphology and misorientation of cross-section of the deposition layer obtained by using wire feeding speed of 4 m/min. (Note: (**a1**,**a2**) MIG, (**b1**,**b2**) M-MIG with magnetic excitation current of 1A, (**c1**,**c2**) M-MIG with magnetic excitation current of 2A, (**d1**,**d2**) M-MIG with magnetic excitation current of 3A, the number 1 and 2 are expressed respectively the grain morphology and misorientation).

**Figure 10 materials-15-07500-f010:**
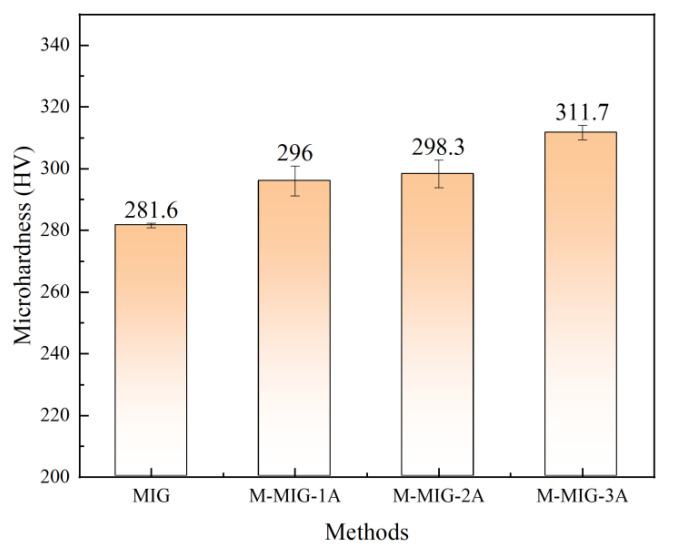
Microhardness of the deposition layer obtained by using wire feeding speed of 4 m/min.

**Table 1 materials-15-07500-t001:** Parameters of Ti-6Al-4V alloy in M-GMAW-AW.

No.	Voltage (V)	Wire Feeding Speed (m/min)	Magnetic Excitation Current (A)
**1**	19	2	0
**2**			1
**3**			2
**4**			3
**5**		3	0
**6**			1
**7**			2
**8**			3
**9**		4	0
**10**			1
**11**			2
**12**			3

## Data Availability

Not applicable.

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
