# Peer review of "Short Circuiting Transfer, Formation, and Microstructure of Ti-6Al-4V Alloy by External Longitudinal Magnetic Field Hybrid Metal Inert Gas Welding Additive Manufacturing"

_materials, 2022, doi:10.3390/ma15217500_

Round 1
Reviewer 1 Report
The authors investigate the external longitudinal magnetic field hybrid metal inert gas welding additive manufacturing method to produce the Ti-6Al-4V alloy part. The effect of process parameters on the droplet transfer formation and microstructure of the part was studied by high speed camera, optical microscope and electron backscattered diffraction. The manuscript is well written and investigate an interesting topic on additive manufacturing. It present enough new results to deserve publication on Materials. I advise the authors to clarify the following points prior to publication:
1) Figures 2 to 4 shows the images of short circuiting transfer for different wire feeding speeds. The authors comment the results on figures 2-4 but there is no comment on the reproducibility of it content.
2) Figure 6 shows a better trace for 4m/min wire feeding speed. What can one expect for higher feeding speed? Is it possible to increase it? Does it mean that 4m/min is better for additive manufacturing this alloy?
In conclusion, this manuscripts requires some minor improvements before publication on Materials.
Author Response
- Figures 2 to 4 shows the images of short circuiting transfer for different wire feeding speeds. The authors comment the results on figures 2-4 but there is no comment on the reproducibility of it content.
A: Thanks for your suggestion. We had modified them.
- Figure 6 shows a better trace for 4m/min wire feeding speed. What can one expect for higher feeding speed? Is it possible to increase it? Does it mean that 4m/min is better for additive manufacturing this alloy?
A: This is a good suggestion. The higher feeding speed also can be used. However, the model of droplet transfer will change from short circuiting transfer to globular transfer or spray transfer. In this work, we mainly focus on short circuiting transfer, therefore, the higher feeding speed is not used.
Reviewer 2 Report
The paper “Short circuiting transfer, formation and microstructure of Ti- 6Al-4V alloy by external longitudinal magnetic field hybrid metal inert gas welding additive manufacturing” by Chao Shi, Hongwei Sun and Jiping Lu is devoted to the study of the influence of the longitudinal magnetic field on the morphology and structure of deposition layer during welding process of Ti-6Al-4V alloy. It is known that magnetic field assisting allows to improve welding process and the investigation of physical principles and peculiarities of this technology is of interest now. So I have no doubt that the subject of the paper is worth to be investigated and it is fitted to Materials. But I do not think that the paper can be published in present form. From my point of view, it should be reworked to make accents on the mechanisms of the physical processes.
Here are some points that should be improved:
- First of all, I would like to see clear explanation or at least some explanation of the influence of magnetic field on the processes discussed in the work.
- I do not think that it is necessary to present Figs.1-4. I think that droplets formation time values presented in Fig. 5 is more than enough. Otherwise the authors should point out on some peculiarities to be distinguished from these Figures.
- As for Figure 6. Can the authors provide some statistics and what are the errors? What is reason of such difference in the influence of magnetic field in different regimes?
- There is no information provided about value of magnetic field near the welding area. The number of coil turns and current is not enough to figure this out. It is also needed some information about welding current values and variation to evaluate the Ampere’s forces acting in the system.
- The part about morphology and structure is presented much better. But presented Figure 8 is totally red and cannot be analyzed.
- From my point of view, the Conclusion part should be totally rewritten with accent on the mechanisms.
Author Response
- First of all, I would like to see clear explanation or at least some explanation of the influence of magnetic field on the processes discussed in the work.
A: Thanks for your suggestion. The influence had been added as follow.
.The liquid metal inside the molten pool had a complex flow under the action of arc pressure, electromagnetic force and other forces. Under the action of strong stirring, a forced convection process would be formed in the weld pool. The high temperature liquid melting pool metal has changing flowing direction under the action of the varying Lorentz force, which made the uniform distribution of temperature field and solute field obtained in molten pool. Therefore, the preferential growth of grain was limited, and the flushing action of the fluid would also promote the formation of new crystal nuclei, which could increases the number of crystal nuclei in the unit area of the liquid molten pool.
- I do not think that it is necessary to present Figs.1-4. I think that droplets formation time values presented in Fig. 5 is more than enough. Otherwise the authors should point out on some peculiarities to be distinguished from these Figures.
A: Thanks for your suggestion. Some difference had been added as follow.
From figures 2-4, it can be concluded that with the different wire feeding speeds, the change tendency of short circuiting transfer were the same. However, the arc shape and droplet size had obviously difference. For the arc shape, the larger area of arc shape obtained by the wire feeding speed of 2m/min due to the instability of welding process. When the wire feeding speed was increased, the stability of welding process was improved which results in the arc shape no obvious change. In addition, the droplet size was reduced by the increase of wire feeding speed.
- As for Figure 6. Can the authors provide some statistics and what are the errors? What is reason of such difference in the influence of magnetic field in different regimes?
A: Thanks for your suggestion. The “spatter” had been indicated in Figure 6. And the reason is given in paper, as follow
Fig.6(a) and (b) shows the surface morphology of the deposition layer with the wire feeding speed of 2m/min. The deposition layers with different processes can not be forming, which consists of discontinuous deposition layer. It can be concluded that the non-forming of deposition layer maybe affected by the instable of droplet transfer (Fig.2). The discontinuous flow of molten metal was the directly reason of discontinuous deposition layer.When the wire feeding speed is increased to 3m/min, the forming of deposition layer obviously changed as shown in Fig.6(c) and (d), however, there are some spatters on the surface of deposition layer due to the change of droplet transfer path caused by magnetic field. In addition, when the magnetic excitation current is increased to above 2A, the forming of deposition layer is poor. This is because that a larger electromagnetic force obtained by electromagnet could adversely affect the droplet transfer, such as great change of arc shape and spatter. When the wire feeding speed of 4m/min is used, the uniform forming of deposition layer is obtained as shown in Fig.6(e) and (f). From Fig.4, it also can be found that the stable droplet transfer was obtained with the wire feeding speed of 4m/min. Therefore, the stable droplet transfer was the key to getting the uniform forming of the deposition layer.
- There is no information provided about value of magnetic field near the welding area. The number of coil turns and current is not enough to figure this out. It is also needed some information about welding current values and variation to evaluate the Ampere’s forces acting in the system.
A: Thanks for your suggestion. When the number of coil turns is constant, the value of magnetic field near the welding area is only affected by the current. In addition, the wire feeding speed of 2m/min,3m/min and 4m/min are respectively corresponded the welding current of 150A, 165A and 180A.
- The part about morphology and structure is presented much better. But presented Figure 8 is totally red and cannot be analyzed.
A: Thanks for your suggestion. The figure had been changed as follow
- From my point of view, the Conclusion part should be totally rewritten with accent on the mechanisms.
A: Thanks for your suggestion. The Conclusion part had been rewritten.
Reviewer 3 Report
Fig. 2-4 are unclear, enlarge them!
Good research work.
Author Response
Fig. 2-4 are unclear, enlarge them! Good research work.
A: Thanks for your suggestion, they had been enlarged.
Round 2
Reviewer 2 Report
The work was improved and all my comments were addressed. I think that present version can be published in Materials
Author Response
A:Thanks for your comment.